# Safety and Family-Centered Care during Restriction of Hospital Visits due to COVID-19: The Experience of Family Members

**DOI:** 10.3390/jpm12101546

**Published:** 2022-09-20

**Authors:** Tânia S. P. Correia, Maria Manuela F. P. S. Martins, Fernando F. Barroso, Olga Valentim, César Fonseca, Manuel Lopes, Lara G. Pinho

**Affiliations:** 1Instituto de Ciências Biomédicas Abel Salazar (ICBAS), Universidade do Porto (UP), 4050-313 Porto, Portugal; 2CINTESIS (Centro de Investigação em Tecnologias e Serviços de Saúde)—NursID (Innovation & Development in Nursing), 4050-313 Porto, Portugal; 3Escola Superior de Saúde Ribeiro Sanches (ERISA)–IPLUSO, 1950-396 Lisboa, Portugal; 4Escola Superior de Enfermagem do Porto (ESEP), 4050-313 Porto, Portugal; 5Centro Hospitalar de Setúbal, 2910-446 Setúbal, Portugal; 6Escola Superior de Enfermagem de Lisboa (ESEL), 1600-096 Lisboa, Portugal; 7Nursing Department, Universidade de Évora, 7000-811 Évora, Portugal; 8Comprehensive Health Research Centre (CHRC), Universidade de Évora, 7000-811 Évora, Portugal

**Keywords:** family nursing, family-centered care, hospitalization, COVID-19, patient safety, safety management

## Abstract

Background: Person and Family Centered Care (PFCC) has demonstrated important contributions to health care outcomes. However, in response to the need for safety due to the pandemic COVID-19, measures were taken to restrict hospital visits. So, the aim of this study was to understand the healthcare experience of family members of patients hospitalized during the pandemic period regarding safety and person- and family-centered care. Methods: Qualitative interpretative study, conducted through semi-structured interviews with six family members of people hospitalized during the pandemic period. Content analysis was performed using Atlas.ti software version 22 (Berlin, Germany) and Bardin’s methodology. Results and Conclusions: Restrictions on hospital visits due to the pandemic of COVID-19 have led to a distancing of families from the hospital setting and influenced healthcare practice, making it difficult to involve families in the care process. In some cases, healthcare professionals made efforts to provide PFCC, attempting to minimize the impact of the visitation restriction. However, there were reported experiences of care delivery that did not consider social and psychological factors and did not place the person and family at the center of the care process, relying instead on the biomedical model. These practices left out important factors for the provision of safe care. It is crucial, even in pandemic settings, that healthcare professionals provide person- and family-centered care to the extent possible, promoting the safety of care. The family should be involved in the care of the person in the inpatient setting.

## 1. Introduction

Concerns and orientations about the humanization of healthcare and the involvement of family members in hospital care have emerged with more focus in the last decades. In this sense, until the emergence of the Corona Virus Disease pandemic (COVID-19), the institutional and health professionals’ practices toward families in this context were improving [1].

Family members can play a key role in the care process during hospitalization. This includes providing stability, emotional support to the hospitalized person, and support in various needs and/or activities [2]. The presence of the family in the hospital context enables the family’s own need for support and information to be met and gives them the opportunity to be close to the hospitalized patient [3].

Person and Family Centered Care (PFCC) is an approach to assessment, planning, and healthcare implementation that is based on beneficial partnerships between clients, families, and healthcare professionals. Such partnerships, at the clinical, strategic, and policy levels, are believed to be essential to ensure quality and safety of care [4,5].

Scientific evidence has shown that when healthcare services administrators, healthcare professionals, patients, and family members work in partnership, the quality and safety of healthcare increase, costs decrease, and patient, family, and healthcare professionals’ satisfaction increase [4].

However, to contain virus transmission, in the context of the COVID-19 pandemic, many health systems adopted measures to restrict visits to hospitalized patients. These measures sought to prioritize patient safety.

Health safety is considered a complex phenomenon due to the involvement of numerous factors, so it is essential to analyze the mechanisms that contribute to errors. James Reason developed the Swiss Cheese model (Figure 1) to explain the dynamics of failures in a system [6]. He considers that systems or organizations have barriers, such as equipment, people, or technology, which are strategically positioned to prevent error. Ideally, the barriers would have no gaps, however, the author recognizes that gaps do exist, and these are represented by the holes in the Swiss cheese slices (barriers). He describes them as dynamic because they open and close in different places. The existence of isolated gaps does not necessarily imply that an error has occurred. An error can occur when failures in all barriers are aligned in a path with risk to the customer’s safety. This model helps risk management by making it more proactive [6,7]

Scientific evidence shows that families make efforts to ensure the safety of their hospitalized relatives. Some of these efforts are in line with recommendations for patient safety [8]. However, families feel unprepared and report a lack of support from health professionals to collaborate in this area [8]. Research findings highlight the need to improve health professionals’ communication with the family about the safety of health care [8].

The World Health Organization (WHO) has published the Final draft of the Global Patient Safety Action Plan 2021–2030 where it presents as one of the strategic goals the involvement of patients and families as partners in safe care [9]. Notes that the safety of health care depends on the involvement of the patient, family members, and caregivers as partners in planning, supervision, fully informed consent, and shared decision-making [9]. The family is considered irreplaceable in the context of care. They know the patient’s health history best, have the potential to collaborate in the observation and surveillance of the patient, are alert to emerging needs, and can be the eyes and ears of the system [9]. For this, it is fundamental to involve and empower families and recognize them as partners in the safety of care [9]. 

During the pandemic period, we found that the nurses’ actions were strongly influenced by the view of the family as a potential disseminator of infection. They assessed family presence more negatively than positively in terms of maintaining safety, an understanding that was not always supported by the available evidence [10,11,12,13].

On the other hand, these professionals identify implications for the patient, family, and care practice of visit restriction measures [3]. The implications of the absence of the family for the hospitalized patient are considered negative in several areas including mental health, resistance to therapeutic adherence, and the feeling of insecurity that jeopardizes the quality and safety of care [3].

Person- and family-centered care (PFCC) is of special importance for the clinical practice of nurses and other health professionals [4]. Along with this recommendation is the need to ensure patient safety. Evidence on the relationship between family involvement and inpatient safety is limited [8]. During the pandemic period, this need gained a strong relevance. So, the aim of this study was to understand the healthcare experience of family members of patients hospitalized during the pandemic period regarding safety and person- and family-centered care.

## 2. Materials and Methods

The removal of families from the hospital environment as a safety measure in a pandemic context is a phenomenon of a complex and multifactorial nature. In this context, we developed a qualitative and interpretative study with thematic analysis, specifically Bardin’s content analysis [14]. 

To answer the aim of this study, we formulated the following research question. How did family members experience the hospitalization of a family member during the pandemic period according to patient safety and the PFCC?

We used the purposive sampling method. The sample consists of relatives of patients hospitalized during the pandemic period of COVID-19 in three hospitals in the North and Centre of Portugal, in internal medicine and general surgery inpatient services. They were selected by convenience, as they were the most accessible and met the pre-established inclusion criteria [15]. The inclusion criteria were: being a family member of a patient hospitalized during the pandemic period of COVID-19; being a family member of an adult patient hospitalized in one of the three hospitals of northern and central Portugal and being available to participate.

Data were collected through individual semi-structured interviews between April 2021 and May 2022 with the aim of knowing the perceptions of family members about the implications of their presence/attendance of their hospitalized sick family members during the COVID-19 pandemic. The participants often answered the questions even before they were asked. Therefore, the semi-structured interview was the best option. The questions in the interview script emerged from the integrative review conducted a phase prior to the study and are as follows [8]:-What visiting arrangements were in place at the hospital during your family member’s hospitalization?-During your family member’s hospitalization, were you allowed to visit him/her?-If yes, in what circumstances, how often, whose initiative and decision was it to do so?-If no, who communicated this decision? Was there justification for this decision?-Were you encouraged to designate a family spokesperson to facilitate effective communication between family members and hospital professionals?-Were you or another family member asked about the health situation prior to the hospitalization (medication, level of autonomy, background, allergies, or other relevant information)? If yes, by whom?-How were you able to monitor your family member’s health status?-Were you able to contact the inpatient service easily? How? Which professionals could you communicate with? What information could you obtain?-Were you able to contact your family member easily? How did you do it?-In this communication with professionals and hospitalized family member, did you feel that your family member was safe? And emotionally, how did you feel?-Did your family member feel safe? And emotionally how do you think your family member felt?-Were you explained the treatment plan for your ill family member during the hospitalization? Do you know what medication your family member was on? If yes, who informed you?-How far in advance were you informed of the discharge? Which health care professional informed you? What information were you given?-Did you feel that there was a preparation process for your family member’s going home? If yes, which health care professional liaised with you and to what extent?-Were you or another family member explained the treatment plan for your ill family member at home? If yes, who informed you?- Were you or another family member explained the treatment plan for your ill family member at home? If yes, who informed you?

Due to the restrictions and fears associated with the pandemic by COVID-19, the interviews were scheduled according to the participants’ preferences and thus conducted via telephone as it was the most accessible to them. In this way, audio recordings were made with the necessary consent. As a consequence of the restrictions on access to hospitals during the pandemic period, we were only able to interview six families. Thus, we were unable to ensure that data saturation was achieved. The six participants are all female, aged between 19 and 77 years (mean age 50.5 years), with educational levels: three participants with a 9th-grade education, one with a 12th-grade degree, one with a bachelor’s degree, and one with a doctorate. Their hospitalized family members were between 84 and 91 years old (mean 86.5 years) and schooling: five hospitalized family members with no schooling and one with a 4th-grade education, with a length of stay between 14 and 35 days (mean 22.8 days). All were discharged home except for one who died during the stay. 

Each interview was transcribed in full, and its transcript was sent to the interviewee for validation. The content analysis was developed with thematic analysis, in three phases defined by Bardin [14]: pre-analysis, exploration, treatment of results, and interpretation. Common contents were searched for in the exploratory analysis, which gave rise to the identification of thematic areas, designated by categories. In coding and categorizing the interviews we used the software Atlas.ti^®^ software version 22 (Berlin, Germany). This software allows you to identify and create the units of analysis (categories) from the significant excerpts of the interviews. The categories were grouped into major thematic areas (families).

The research team included four female and three male members; five Ph.D. professors, a master’s degree, and a nurse specialist with management functions; with specialization qualifications in mental health nursing (4), rehabilitation nursing (1), and community health nursing (1); with post-graduate training in clinical supervision and management of nursing services; with professional experience at the academic and clinical level in several areas, namely at the hospital level in Mental Health and Psychiatry services, Outpatient, Rehabilitation, Emergency, Medicine, Surgery, Orthopedics internments; at the primary health care level and in health management.

Ethical and legal principles were followed. The study was approved by the joint Ethics Committee of the Centro Hospitalar e Universitário do Porto and Instituto Ciências Biomédicas Abel Salazar (ICBAS) of the Universidade do Porto (UP). All procedures were carried out with participants respecting anonymity, confidentiality, and informed consent, as well as the Helsinki Declaration of Human Rights.

## 3. Results

From the analysis of the data obtained in the interviews with family members, 23 categories (codes) were identified, which were grouped into two major thematic areas (families) according to the nature of the experiences described by family members: PFCC and Biomedical Model (Figure 2 and Figure 3). In each family, the safety categories were identified according to James Reason’s Risk Model (2001), namely Safety Barriers and Safety Gaps.

The PFCC involves care in a mutually beneficial partnership between families and health professionals. There are 10 categories in PFCC: *Family well-being; Patient well-being; Easy communication between family and health team; Easy communication between family and patient; Family contribution to the patient’s recovery process; Flexible visiting policy; Identification of family member of reference; Perception of safety of care; Discharge planning with the family; and Therapeutic plan known by the family* (Figure 2). Among these, there are categories that, according to the James Reason Risk Model (2001), can be considered safety barriers: *Easy communication between family and health team; Easy communication family-patient; Family contribution to the patient’s recovery process; Identification of family member of reference; Discharge planning with the family;* and *Therapeutic plan known by the family*.

The Biomedical Model of health and disease has the assumption that all health-disease conditions are explained by physiological abnormalities. This model does not consider social and psychological factors as relevant [16]. A total of 12 categories were identified in the biomedical model: *Total restriction of visits; Impaired patient-family communication; Patient mental health at risk; Patient appeal for discharge; Family distress; Unknown therapeutic plan by the family; No discharge planning with the family; Risk of post-discharge medication errors; Impaired health care team-family communication; Perception of insecurity of care; No identification of family member of reference; Death process without the family* (Figure 3). According to James Reason’s Risk Model (2001), categories compatible with Safety Failures were identified, namely: *Total visit restriction; Compromised patient-family communication; Mental health of the patient at risk; Patient appeal for discharge; Therapeutic plan unknown to the family; Absence of discharge planning with the family; Risk of post-discharge medication errors; Compromised health care team-family communication; No identification of family member.*

The data collection experience was emotionally enriching. It was possible to perceive that the participating family members needed to talk about their experiences. In the experiences described as more negative, they demonstrated the need to feel that someone would listen to them.

## 4. Discussion

Despite the restriction of visits, we were able to identify some congruent practices with the PFCC, which indicates that it is possible to maintain some practices focused on the patient and the family, even at a distance. However, it should be noted that there are factors that can only be overcome with a physical presence, such as the direct provision of care by the family or the intensity of the effects.

### 4.1. Person- and Family-Centered Care (PFCC)

Relatives of hospitalized patients mentioned that the presence of the family next to their relative contributes to the *Family’s well-being*: “The presence of the family is very important, fundamental. (…) And for the family too, we are calmer, more rested because we can see with our own eyes how our relative really is. It’s good for everyone” (E6). The provision of care involving the family and facilitating their presence and collaboration in the hospital setting contributes to the balance of family health [2,17].

Similarly, relatives consider that their presence contributes to the *Well-being of the patient*: “I think so. I think it was very, very, very beneficial, yes. Even though she was very sleepy, maybe because of the medication, or even because of the health problem, I don’t know. I think it was very good when she would open her eyes, to see someone she knew at the foot of her bed. I think it was very very very good for her” (E1). Some participants refer that it is even fundamental for the well-being of the patient: “The presence of the family is very very important, fundamental. A person who is hospitalized is better off with the family around, more peaceful, it’s a comfort” (E6). The available evidence has been showing benefits of family involvement in the context of care, namely by leading to improved patient experience and increased levels of satisfaction [4,18].

Within the PFCC, even with the restriction of visits, participants reported experiences of *Easy communication between family and health team*: “It was very easy. At that time they weren’t that quick to take the call, but you have to understand the time we’re in. But they answered and went on to the service without much trouble” (E1). They also mentioned that communication was possible with different elements of the family: “It was more my nieces who called the service, and they contacted the doctor, then towards the end, the mother gave my name and the physiatrist then started talking to me” (E3). This category has special value because communication is one of the fundamental bases for PFCC and for the safety of care [4,9]. 

With the implementation of visitation restriction, there were participants who mentioned an *Easy family-patient communication*: “Yes, every day we called many times a day. My mother always answered” (E3). There were also reports of nursing intervention in facilitating this communication: “…the first day she was admitted there she didn’t have her belongings with her yet, a nurse made a call with us. We were able to talk to her anyway” (E4). As an alternative to physical presence, there are communication strategies, such as phone calls and video calls, which have shown benefits for the patient such as the reduction of incidence of delirium during hospitalization [18].

Several participants consider the *Family’s contribution to the recovery process* of the hospitalized patient as important: “In recovery, I think it is essential to have the family around” (E1). In addition to presence, they identify the need for involvement in care: “The affections and relational bonds are essential to the recovery process. The family should be involved in the care of the inpatient. If it is the primary caregiver, even more so, because if the caregiver is going to provide care at home, there should be a prior preparation and assessment of the caregiver’s learning needs in terms of care” (P5). This finding is in line with the available evidence considering the advantages and positive results of the implementation of family-centered care for the safety and quality of the health care provided, such as the decrease in readmissions [8,18,19,20,21].

Despite guidelines to restrict visits, *Experiences of flexibility in the visitation policy* were reported: “… we managed exceptional visits. I don’t know if it was because we were sorry, I don’t know if it was because the hospital saw fit, because of the severity of the problem…. I don’t know” (E2). This flexibility was received with satisfaction by the families: “There were no visits, my niece went there twice a week to take her clothes and whatever she asked for, and a few things to eat because she never liked the hospital food… then my niece managed to arrange one visit in the middle of the hospitalization. She and I went there. She talked to the head nurse and thank God I got a visit. It was our initiative. Also because of the length of hospitalization. I believe that after 1 month or so of being there that everyone was entitled to a little visit” (E3). Although this family had only one visit after a month of hospitalization, this one visit was considered very important. Only these two families had the opportunity to visit their relatives. Evaluating each situation in particular and deciding accordingly demonstrates a person- and family-centered approach and in accordance with the emerging evidence that the presence of the family next to the hospitalized person positively influences the results for the patient and for the care process [11,22,23].

The *Identification of a family member of reference* is considered a safe and effective communication strategy with the family and was mentioned by the participants: “Yes, at the time my aunt’s contact remained. They said that the spokesperson was for when they must communicate some decision, to call that person” (E4). According to studies, the better the communication with the family, in addition to improving safety outcomes, the better their perception of the quality and safety of health care [24]. These data corroborate the category identified in this study: *Perception of safety of care*. In this context, the participants mentioned that the presence of the family is essential for patient safety: “The person is safer with the family around, because of their presence and because they ask questions. (E1). That the fact that they show concern influences health professionals for the safety of the patient: “… the family has a role. It shows health professionals that there is interest in the patient and that he has someone who cares and accompanies him. In case they forget and think that the patient is being deposited there and is on their own” (E4).

*Discharge planning with the family* is one of the strategic elements for family-centered intervention with the goal of patient safety in the period after discharge [24]. Most likely due to the restriction of visits, only one of the participants reported this planning involving the family: “For the mother to leave, it was me who took care of it,… meanwhile I was doing construction work in my mother’s house, and I told the physiatrist about it, and she asked me how the construction work was, if it was already finished. They were finished on Saturday, and mom was discharged on Monday. It was all with the physiatrist, I actually liked to meet her… but if I told the physiatrist that I had no conditions to take care of my mother, they wouldn’t send her away, they would keep her there” (E3). In this case, there was coordination between the hospital and the family to adjust the timing of discharge, although there was no direct involvement in the care.

Another important strategy for patient safety is the involvement of the patient, family members, and caregivers in partnership in the planning, supervision, fully informed consent, and shared decision-making of healthcare [9]. This strategy was identified in the speech of one of the participants: “Not the medication, but they explained to us what the treatment plan was, what they were going to treat and do” (I4). Although not all the information is given, we consider the category *Therapeutic plan known to the family*. Even so, it is necessary to evolve in the sense of including the family in the planning process and not just making them aware of it.

According to the analysis of the speeches of family members of people hospitalized during the pandemic period, it is possible to verify that, despite the measures to restrict visits, the PFCC continued to be developed. In this regard, adaptations were necessary, which demonstrate the commitment of the health teams to the well-being of the hospitalized person. 

From the safety perspective, it is possible to verify that the practices identified as PFCC have the potential to contribute to the safety of health care. The categories *Easy communication between family and health team; Easy family-to-patient communication; Family contribution to the patient’s recovery process; Identification of a family member of reference; Perception of safety of care; Discharge planning with the family; and Therapeutic plan known by the family;* can be considered as safety barriers according to James Reason’s Risk Model (2021). Therefore, they should be included as strategies to promote the safety of care in partnership with families in hospital settings.

### 4.2. Biomedical Model

*Total restriction of visits* represents an important barrier to PFCC by its distance from the care process but also demonstrates inflexibility and lack of individualized assessment to adapt practices in health services: “Zero policy. They didn’t allow any visits. My aunt even insisted, but even though I called every day, I didn’t insist, it wasn’t worth it. It’s as if he had been abandoned in the hospital. He wasn’t, but it’s as if he had been” (E6).

While some participants reported that the health team acted as a facilitator of communication with the patient, others reported *Compromised family-patient communication*: “My father had his cell phone with him. Although he didn’t make calls, he answered. We couldn’t have a dialogue with him, he was very tired, he would just answer that he was fine and we would send kisses. When he got worse, towards the end, he wouldn’t answer, we only knew what they told us… I was never allowed to go visit my father, even when he got worse, he wouldn’t even answer his cell phone, we only knew what they told us… we couldn’t really see how he was doing” (E6). 

The instability generated by hospitalization added to health changes is very disturbing. If we add the impediment of contact with significant family members, we may be facing very difficult experiences with an impact on the mental health of the hospitalized person [21]. *Mental health of the patient at risk* was mentioned by some participants as a consequence of the absence of visits: “People get depressed, when they are there alone, I think, they get depressed besides the disease which, in itself is not good, I think they get depressed, yes. And that doesn’t help at all, it’s not good” (E5). Implications of the restriction of visits on the mental health of hospitalized patients were also reported by nurses, namely sadness, depressed mood, anxiety, isolation, loneliness, confusion, and agitation [3].

Due to the lack of visits, some participants reported that their hospitalized family members presented *Patient’s appeal for discharge*: “After a week of hospitalization he began to insist to be discharged, he missed his family… after a month of hospitalization in the same hospital my grandmother requested discharge because she could not stand without visits to wait any longer for a vacancy in the convalescent unit for rehabilitation” (E5). This rush to discharge can compromise proper discharge planning and affect the recovery process and even contribute to adverse events at home [25].

The category *Family distress* was identified in different participants’ speeches due to the impossibility of visiting their hospitalized relative. Such remoteness was reported as a reason for great distress and suffering for the whole family and was not taken into consideration by the health team: “It was driving us crazy, we were all very worried about not being able to see him. It was a very big concern…. He even called my mother, I still haven’t asked her today what they talked about… maybe he asked to pick him up or see him, or maybe he said goodbye, I don’t know. It was very hard, it still is. We knew he wasn’t well, but we didn’t expect him to pass away, much less in such a short time, for that we were not prepared. Sometimes I think if we had been there, if it wouldn’t have been different, I don’t know, maybe he would have died anyway… I don’t know.” (P6). In this situation, suffering is perpetuated after hospitalization. It is important to consider that the consequences of restrictive measures to hospital visits may perpetuate beyond the time of hospitalization. Several studies have presented consonant results, the policies of visit restriction have repercussions on the well-being and mental health of family members [17].

The *Family’s unawareness of the therapeutic plan* in addition to revealing the exclusion of the family from the decision-making process implies the lack of informed consent of the family of disoriented patients for decisions about the care to be provided: “We were not informed about the treatment during hospitalization” (E1). This practice does not comply with the WHO final draft Global Patient Safety Action Plan 2021–2030 guidelines [9] for family involvement in partnership for safer care.

The *Absence of discharge planning with the family* was also frequently reported by most of the participants: “There was no discharge preparation by any professional, because who came downstairs were two Nursing Assistants who only gave us the mother and that was it. They brought a sealed letter to the doctor, and a sealed letter to the nurse, but nobody at the hospital said anything to us. All they did was bring my mother down here and give her to us. It wasn’t until my mother was discharged that she came with the medical information and that’s when we knew what we had to do” (E1); “they contacted us with the possibility of my grandmother being discharged the next day. And they asked if we had conditions, we didn’t ask what kind of conditions we would have to have. We said yes” (E4). They also revealed that little information about discharge was given by the medical team: “Explanations may have been given to my niece Maria (fictitious name) by a doctor but in that service, in the nursing part I didn’t feel any of that… The only nurse who talked to me about it was a friend of the family and who was not from the service” (E2). As mentioned earlier, discharge planning is critical to ensure patient safety. When this does not happen, adverse events may arise. The most reported with evidence are medication errors [25], a risk also mentioned by one of the participants and that originated the subcategory *Risk of medication errors on post-discharge*: “As far as I know, they only explained to my grandmother. Regarding the medication she and my aunt were messing up with the medication and contacted the family doctor” (E5).

*Communication Health team-Family engagement* was very present in the participants’ speeches: “The doctor, for example, could have called a son, could have talked to him personally. They could have arranged to talk to one of the sons. Talk more about the mother’s health condition. Because it was never anything like that. One person would call and know one thing, then another person would call and know another” (E1). On admission to the emergency room, there were also communication problems: “When I took my mother to the hospital, another sister-in-law of mine went with me…she went to do the chart…they put a phone number of another sister of mine. There are a lot of us. And this sister of mine, she wasn’t there at the time and when the information came that the doctor would call family member that she was out here when she had something to report. That he was going to call on the phone…I don’t know if the doctor actually called him to find out about some medication that my mother was already taking…I went with my mother to the triage, from there on in, no more. She was already alone, which is also a very bad thing. From 3 pm until about midnight… there was very little dialogue…” (E2). They also mentioned errors in the information given: “There were some lapses, sometimes even confusion of information, sometimes they gave switched information. For example, my grandmother even had to go to a hospital in Porto to have a prosthesis put in, I think, and one time we called there and they said she was in Guimarães having a different examination, on her chest or something like that” (E4). 

One participant privileged communication with the doctor and devalued nursing information: “I was able to keep up with my father’s health status because I called every day to find out information from the medical team… I usually talked to the medical team during the week. On the weekend I would talk to the nursing team…they would just say everything was fine and then it wasn’t anything like that, so it was more with the medical team” (E6). Results of studies conducted in institutions with open visiting policies show improvements in the communication between the family and the health team as well as the trust between both [20]. In this sense, the participants referred to a Perception of insecurity of care by not being able to be present in the hospital: “… the process of recovery and rehabilitation in an unknown environment and without people belonging to their ties is time-consuming and becomes a risk to mental and physical health, since both are linked” (E5).

In addition to the previously mentioned difficulty in communicating with the health team, the participants mentioned that there was *no use of the strategy of identifying the relative of reference*: “…in the end, just before my father passed away, it was very bad because they called my mother at 9 pm, she who is 91 years old, to say that my father was very ill and to expect the worst. I was going to sleep with her those days, but at that time I was still organizing my things at home when my brother, who lives next door to her, calls me, very distressed. She couldn’t even say things right, but she was saying that my father was going to die. It was very difficult. We had asked not to call her, to call me if something happened” (E6). This event jeopardized the health and safety of the elderly person who received the information. It reveals a lack of knowledge or devaluation of the information given by the daughter to the health team. In addition to being insensitive to the family’s needs, it is unsafe and is a dehumanized care practice that can bring harm to the family. Once again it is shown that humanization of health care is a fundamental imperative that has a very close relationship with PFCC. It also includes a personalized, holistic view of the person and considers that the affective relationships of patients should be taken into consideration [26]. 

Another category that is not in line with the humanization of care is the *Death Process without the family* reported by one of the participants as a direct consequence of the policy of restriction of visits: “It is very hard to have a family member hospitalized and not be able to see him. In my father’s case, even worse, he passed away without seeing us. It’s still very hard for me to talk and think about it, what he thought or felt, I don’t know… he even asked my aunt to go and get him and he asked me to go and see him. I said we would see each other later. It didn’t happen anymore” (E6). This practice, besides being dehumanized, does not respect evidence-based practice. The importance of the role of families in the care of terminally ill patients is widely recognized. It is also known that care should include the patient and the family before and after the death of the patient [27]. 

These identified categories were encompassed within the Biomedical Model as they represented practices that did not consider social and psychological factors as integral to health care practice. Resulting from this insensitivity to these factors, experiences of potentially unsafe and dehumanized care were described by family members. 

From a security point of view, we can identify these categories as potential failures in the security of healthcare systems. These categories are: *Total visit restriction*; *Family-patient communication compromised*; *Mental health of patient at risk*; *Patient appeal for discharge*; *Therapeutic plan unknown to family*; *No discharge planning with family*; *Risk of medication errors post-discharge*; *Health care team-family communication compromised*; *No identification of reference family member*. These should be known and identified in the health systems to develop strategies to avoid these failures.

The limitation of this study is the small number of participants, so it should be replicated with a larger number of participants to ensure data saturation. Since this is a qualitative study, it allows us to know the dimensions and nature of the phenomenon under study, but not its extent and impact. Thus, we suggest that quantitative studies be conducted in this area to understand the extent and impact of the phenomenon under study.

## 5. Conclusions

The restrictions on hospital visits due to the COVID-19 pandemic led to a distancing of families from the hospital environment and influenced healthcare practice, making it difficult to involve families in the care process. In this sense, data analysis demonstrates a greater number of categories identified in the Biomedical Model compared to the categories identified in the PFCC. These data demonstrate that these restrictive policies may have influenced the provision of healthcare in the biomedical sense to the detriment of care oriented toward family involvement. 

Still, even with restricted visits, it was possible to identify, through the family members’ speeches, that some health professionals made efforts and adaptations for the development of practices compatible with the PFCC. These data demonstrate that the health teams maintained a commitment to the well-being of the hospitalized person. In this context, we identified experiences of PFCC with the potential to contribute to patient safety such as effective communication with the family, family involvement in the process of recovery and care planning as well as discharge and identification of the family member of reference. 

On the other hand, the care experiences described by family members that did not consider social and psychological factors did not place the person and the family at the center of the care process. Thus, they were grouped under the biomedical model. As a result of these practices, family members described experiences of care with potential risks to the safety of patients and family as well as dehumanization of care. Thus, the total restriction of visits; problems in communication between the family, the health team, and the hospitalized person; the appeal of the patient for discharge; the lack of articulation with the family in therapeutic and discharge planning; and the non-identification of a family member of reference, are potential failures of safety in health systems and should be identified and avoided. 

It is possible to verify that practices based on the biomedical model leave out important factors for the safety of care. In contrast, PFCC is related to care practices with the potential to promote health care safety. 

The restrictions on hospital visits may have driven a major setback in the way health professionals develop their relationships with families in hospital settings. This repercussion may go beyond the time in which these restrictions are in force, both for future care practice and for patients and families. 

Given the results of this study, we recommend continuing to conduct research studies in this area, to detect possible flaws in hospital policies and to be able to correct them to contribute to the practice of PFCC oriented to patient safety.

## Figures and Tables

**Figure 1 jpm-12-01546-f001:**
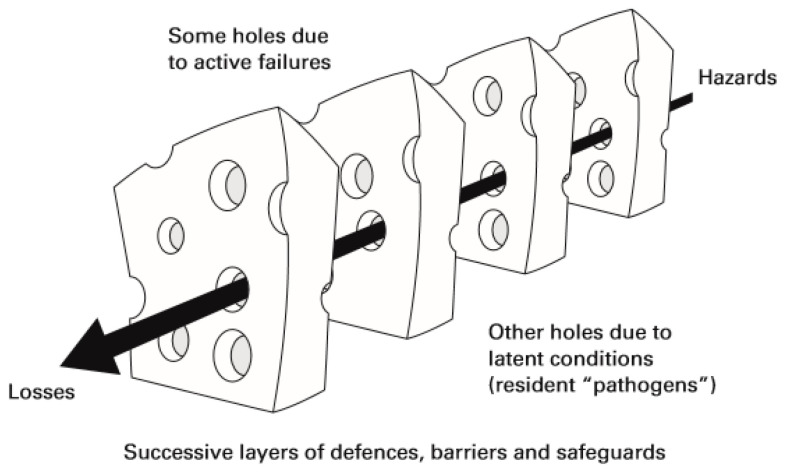
“Swiss Cheese” risk model of James Reason [7]. Source: Reason, J.T.; Carthey, J.; Leval, M.R. Diagnosing “vulnerable system syndrome”: An essential prerequisite to effective risk management. *Qual. Health Care*
**2001**, *10*, ii21–ii25.

**Figure 2 jpm-12-01546-f002:**
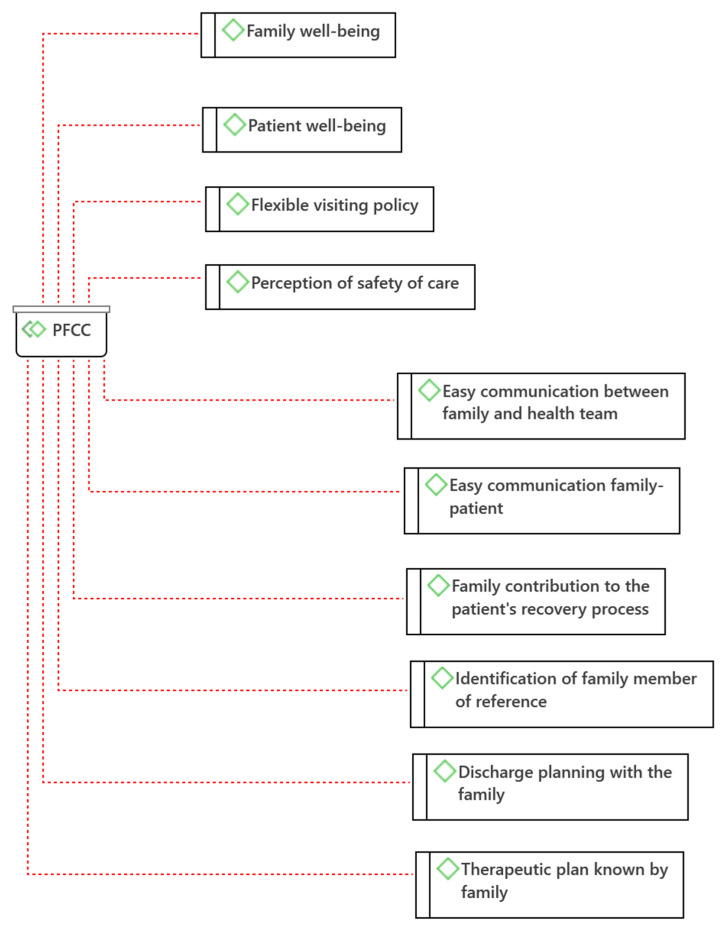
Person- and Family-Centered Care (PFCC) Categories (Atlas.ti^®^).

**Figure 3 jpm-12-01546-f003:**
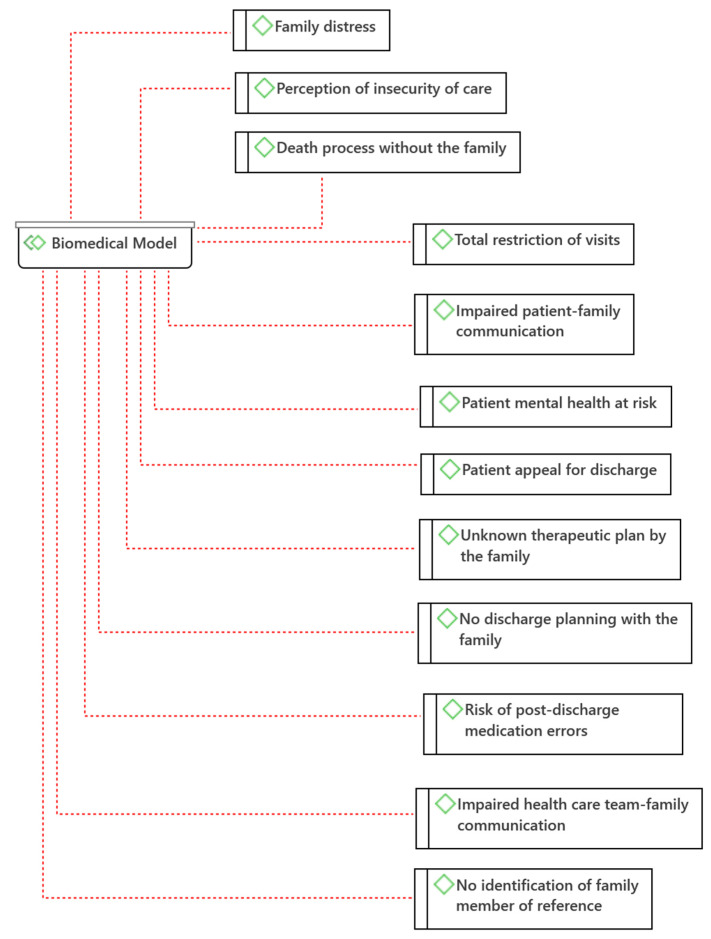
Biomedical Model Categories (Atlas.ti^®^).

## Data Availability

Not applicable.

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
