# Peer review of "Safety and Family-Centered Care during Restriction of Hospital Visits due to COVID-19: The Experience of Family Members"

_jpm, 2022, doi:10.3390/jpm12101546_

Round 1

Reviewer 1 Report

I have read this study with interest and appreciate the opportunity to review it. However, I going to give recommendations to improve this proposal.

Related to the research team and reflexivity: authors should provide some information about Personal Characteristics of the team (e.g., credentials, occupation, gender and experience and training).

In addition, what methodological orientation was stated to underpin the study? (e.g., Grounded theory, discourse analysis, ethnography, phenomenology, content analysis).

Related to interview guide, were questions, prompts, guides provided by the authors? Was it pilot Tested?

Limitation study should be improve because they are very poor.

Author Response

Dear Reviewer!

First of all, we would like to thank you for your comments and suggestions for improving this work, to which we will respond below:

Point 1: Related to the research team and reflexivity: authors should provide some information about Personal Characteristics of the team (e.g., credentials, occupation, gender and experience and training).

 Response 1: Thank you very much for the suggestion. We have added information about the research team composition in lines 189 to 196.

Point 2: In addition, what methodological orientation was stated to underpin the study? (e.g., Grounded theory, discourse analysis, ethnography, phenomenology, content analysis).

 Response 2: Once again we thank you for your comment. Indeed, this is a study whose methodological orientation is based on content analysis (lines 126,127).

Point 3: Related to interview guide, were questions, prompts, guides provided by the authors? Was it pilot Tested?

 Response 3: The script questions emerged from the integrative review conducted a phase prior to the study (lines 142-144).

Point 4: Limitation study should be improve because they are very poor.

Response 4: Thank you again for your comment. We further elaborate on the limitations of the study and suggestions for future research (lines 501 - 504).

Once again we thank you for your suggestions that contributed to improve this paper.

Best regards 

Reviewer 2 Report

The research study provides valuable experiences of families with the hospitalization of their relatives during the Covid pandemic, despite the low number of family members involved. I positively evaluate the use of the comparison of the PFCC model and the biomedical model. The content analysis procedure was based on the phases according to Bardin. Thematic areas and categories were the result of processing using Atlas.ti® software, which resulted in categories related to PFCCand the Biomedical Model, which is clearly recorded in the diagrams. The results and discussion are processed in a logical and explanatory manner.

I have some comments that should be considered:

1. In the methodology of qualitative and interpretative thematic analysis used, it would be appropriate to describe how the interviews were conducted (face to face?, where? how was the interview recorded, what else was important for recording?)  The methodology needs more rigour.

2. The questions listed for the semi-structured interview were numerous and quite guiding.
3. On the formal side, there are only minor errors that are on lines:  17 (align);  435 (Neste âmbito também referiram erros na informação dada: "-translate into English);  592 (it isn´t clear what means nomber 102763); 594 (align) and in some bibliographic reference is marked p as a side and somewhere not.

I consider the research study useful and suitable for publication with minor corrections.

Author Response

Dear Reviewer!

First of all, we would like to express our gratitude for the positive comments that motivate us to continue working towards some contribution to improving the quality and safety of health care. We also thank you for the suggestions that allowed us to improve this work, which we will respond below:

  1. In the methodology of qualitative and interpretative thematic analysis used, it would be appropriate to describe how the interviews were conducted (face to face?, where? how was the interview recorded, what else was important for recording?)  The methodology needs more rigour.

 Response 1: Thank you for your comment. Effectively the interviews were scheduled and conducted according to the preferences of the participants and therefore conducted via telephone and the respective audio recorded with due consent (lines 175 - 178)

  1. The questions listed for the semi-structured interview were numerous and quite guiding.
    Response 2: The questions in the interview script emerged from a literature review conducted in the previous phase of the study. We did not feel that the questions influenced the responses of the participants, who often responded to the questions even before they were asked. Therefore, the semi-structured interview was the best option (lines 142-145).

  1. On the formal side, there are only minor errors that are on lines:  17 (align);  435 (Neste âmbito também referiram erros na informação dada: "-translate into English);  592 (it isn´t clear what means nomber 102763); 594 (align) and in some bibliographic reference is marked p as a side and somewhere not.

 Response 3: All the faults identified on these lines have been corrected.

Once again we thank you for your suggestions that contributed to improve this paper.

Best regards 

Round 2

Reviewer 1 Report

The manuscript has been improved and can be published